# Comparison of ANN and ANFIS Models for AF Diagnosis Using RR Irregularities

Suttirak Duangburong [1], Busaba Phruksaphanrat [1,*] and Sombat Muengtaweepongsa [2]

1 Research Unit in Industrial Statistics and Operational Research, Industrial Engineering Department, Faculty of Engineering, Thammasat School of Engineering, Thammasat University, Pathum Thani 12121, Thailand
2 Center of Excellence in Stroke, Faculty of Medicine, Thammasat University, Pathum Thani 10121, Thailand
* Correspondence: lbusaba@engr.tu.ac.th

**Abstract:** Classification of normal sinus rhythm (NSR), paroxysmal atrial fibrillation (PAF), and persistent atrial fibrillation (AF) is crucial in order to diagnose and effectively plan treatment for patients. Current classification models were primarily developed by electrocardiogram (ECG) signal databases, which may be unsuitable for local patients. Therefore, this research collected ECG signals from 60 local Thai patients (age $52.53 \pm 23.92$) to create a classification model. The coefficient of variance (CV), the median absolute deviation (MAD), and the root mean square of the successive differences (RMSSD) are ordinary feature variables of RR irregularities used by existing models. The square of average variation (SAV) is a newly proposed feature that extracts from the irregularity of RR intervals. All variables were found to be statistically different using ANOVA tests and Tukey's method with a p-value less than 0.05. The methods of artificial neural network (ANN) and adaptive neuro-fuzzy inference system (ANFIS) were also tested and compared to find the best classification model. Finally, SAV showed the best performance using the ANFIS model with trapezoidal membership function, having the highest system accuracy (ACC) at 89.33%, sensitivity (SE), specificity (SP), and positive predictivity (PPR) for NSR at 100.00%, 94.00%, and 89.29%, PAF at 88.00%, 90.57%, and 81.48%, and AF at 80.00%, 96.00%, and 90.91%, respectively.

**Keywords:** atrial fibrillation (AF); ANN (artificial neural network); ANFIS (adaptive neuro-fuzzy inference system); RR interval irregularity; SAV (square of average variation)

## 1. Introduction

The most common cardiac arrhythmia is atrial fibrillation (AF), and diagnosing and classifying its symptoms with a non-invasive procedure is challenging. AF patients have a high risk of heart disease and stroke. Compared to the overall population, Thai AF patients have a prevalence rate of 5 in 100,000. Ischemic stroke patients have a 1.5–1.9 higher chance of having AF compared to a healthy person in the same age group. About 5–21% of acute ischemic stroke patients have AF, but generally do not realize it, which is very serious [1]. If there were a medical tool or a calculation method to help doctors diagnose AF symptoms or risks from the beginning with acceptable time intervals, then the rate of death could be reduced. Pulse palpation, photoplethysmography, oscillometric blood pressure, and electrocardiogram (ECG) are typical methods of AF investigation, while ECG is the essential physiology equipment used to show bio-signals influenced by an alteration in cardiac autonomic tone. In a general examination, only one page of multichannel ECG signals with approximated 10-s recordings is shown, and abnormal symptoms usually do not appear. Abnormal heartbeat patterns in AF patients cause abnormal electrical signals in the atrial chamber of the circulatory system [2]. A total of 24 h of ECG and heart rate indicators are recommended in the guidelines for AF diagnosis [3]. Figure 1 shows examples of NSR, PAF, and AF signals.

Most conventional ECG devices in hospitals that detect AF symptoms using RR interval signals have limitations in accuracy. The typical waveform comprises the P wave,

QRS complex, and T wave. However, when irregular RR intervals occur, the absence of distinct repeating P waves, fluctuating P waves (f-waves), irregular heart rate, and inconsistent heart rate variability (HRV) are the signs of AF [4,5]. ECG signals of AF patients are not consistent and are nonlinear, so they create difficulty in the judgment of AF patients. Investigations of the relationship between the number of reentrant waves in the atrial chambers and abnormal AF signals from ECG signal patterns using signal processing techniques in ECG analysis detected the absence of P waves or RR interval irregularities [6], thus confirming AF patients through long-time ECG recordings. However, P wave and RR intervals are always contaminated with noise artifacts, making it hard to accurately detect AF signals based on these morphological characteristics [7].

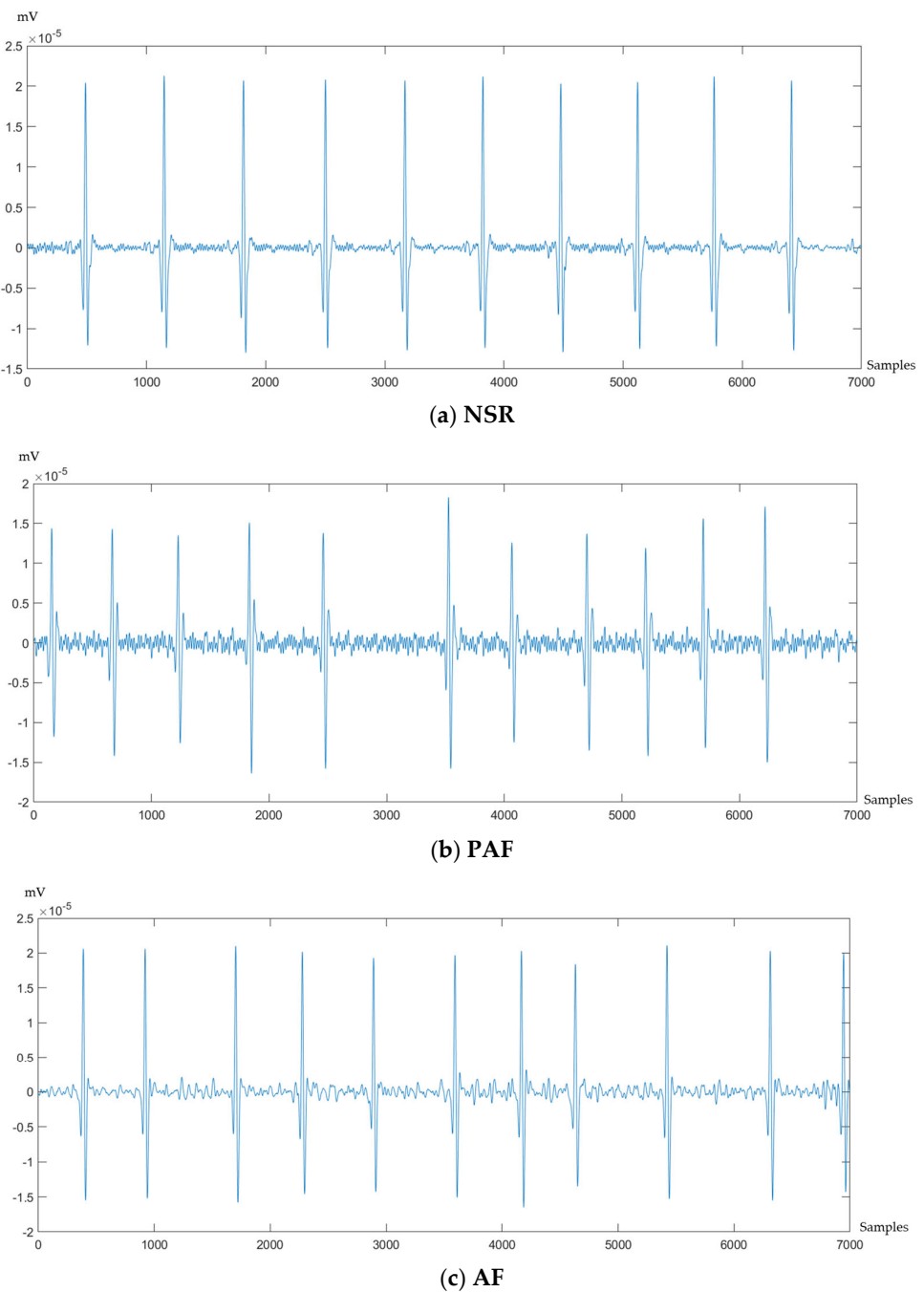

**Figure 1.** Examples of ECG segments for three types of signals.

ECG decomposition is P-QRS-T pattern detection, which requires noise reduction and base wander removal. The characteristics and location of the wave were compared with the standard morphological values of the ECG signals. Time-domain and frequency-domain analysis, commonly used in ECG decomposition in most existing research work for abnormality classification, were both used to extract different kinds of information, and their combination gained more knowledge from the signals. ECG classification methods of AF detection also use feature extractions. The main features from the independent component analysis have been used in many pieces of research to classify the abnormality of patients. Previous research shows that RR interval irregularity can help AF diagnosis using the coefficient of variance (CV) [8], the root mean square of the successive differences (RMSSD) [9], and the median absolute deviation (MAD) [10].

Computer-aided medical diagnosis (CAMD) is a medical procedure to help diagnose quickly and increase the efficiency of services [11]. Mathematical models developed by learning and gaining intelligence with experience were used to construct CAMD in many applications [12]. The artificial intelligence (AI) methods closely relate to the statistics of ECG. The ability of CAMD to diagnose in the medical field requires significant variables in learning and many computational techniques [13]. Machine learning has become widely used in clinical practices, especially with ECG data, such as in support vector machine (SVM), K-nearest neighbor (KNN), and decision tree (DT) analyses [14]. Heart abnormality classification was the most common application of ECG data [15]. Artificial neural networks (ANN) and adaptive neuro-fuzzy inference systems (ANFIS) are currently used to construct models for many applications, such as cancer diagnosis, cardiovascular disease classification, risk assessment for hypertension, atherosclerosis level classification, and predicting symptoms of AF patients [16]. ANN and ANFIS are also powerful classification methods in many applications [17].

Deep learning has become a widespread technique and is most commonly used in supervised machine learning problems. It did not require the detection of P and R peaks or feature designs for classification [18]. However, it can perform well for big datasets [15]. Ebrahimi et al. (2020) showed the list of deep learning techniques for ECG arrhythmia classification from 75 studies and found that the convolutional neural network (CNN) was the dominant technique [19]. They also found that deep learning sped up the process, but the results were not exceptional for finding an abnormal heartbeat. Moreover, deep learning requires an extensively large number of datasets. Feature designs for classification should be used if the number of datasets is not broadly large. Efficient variables should be investigated. As a result, doctors can obtain more information to help with making the right decision for treatment, which can increase the chance of patients returning to their normal physical condition as soon as possible.

Most AF classification models were developed from the MIT-BIH Atrial Fibrillation database [20–22] to compare the performance of computational methods suitable for medical applications. The diagnosis of AF uses feature extraction to find significant variables, including the function algorithm, the time-varying coherence function, conditional entropy, frequency domain tracking, and deep learning [23]. These feature extraction characterizations are used and provide different performance values depending on the suitability of the measuring device and its practical application. Datasets of databases have been used as inputs for the classification models that can obtain high accuracy. However, it may not be suitable for the current population of local patients because the database has been developing for a long time. Therefore, a classification model for local patients is necessary for physicians to analyze AF patients more accurately.

The novelty presented in this research is the newly proposed feature, the square of average variation (SAV), for the classification of NSR, PAF, and persistent AF using patients' data that outperforms the others. The classification of these three types of AF categories has not been investigated. ANN and ANFIS methods were employed with different parameters to find the best classification model. To the best of our knowledge, ANN and ANFIS have not been applied to classify NSR, PAF, and persistent AF with

local patient datasets. The proposed classification model and the newly proposed feature (SAV) show outstanding high performance in accuracy, sensitivity, specificity, and positive predictivity in this research.

This research aims to propose a classification model that is practical for local patients and efficient RR irregularity variables. The current RR irregularity variables and a newly proposed feature SAV for the classification of NSR, PAF, and persistent AF were examined. Data from local patients were utilized instead of using datasets from databases, as other papers do, which is more practical. Comparing methods with different datasets cannot guarantee the efficiency of the methods. So, this research compared ANN and ANFIS using the same datasets from Thai patients. This paper addresses the following research questions:

1. What is the best RR irregularities feature for AF classification?
2. Among ANN and ANFIS techniques, which one is better for AF classification for local patient datasets?
3. What are the best parameter settings for ANN and ANFIS for constructing the AF classification models?
4. How is the performance of the proposed models based on ANN and ANFIS?

## 2. Materials and Methods

This research extracted significant features for fitting the classification model of NSR, PAF, and AF signals based on the procedures in Figure 2. First, the data were collected from ECG recordings in a hospital in Thailand. Then, data preprocessing removed noise and adjusted base wander. Next, the wave components were decomposed using the continuous wavelet transform (CWT) and calculated feature extraction using RR irregularities. Next, feature selection found significant variables for the classification model using a statistical test. Then, ANN and ANFIS techniques were applied to train and test datasets. Finally, performance evaluation was compared among significant variables to find the best AF diagnosis model.

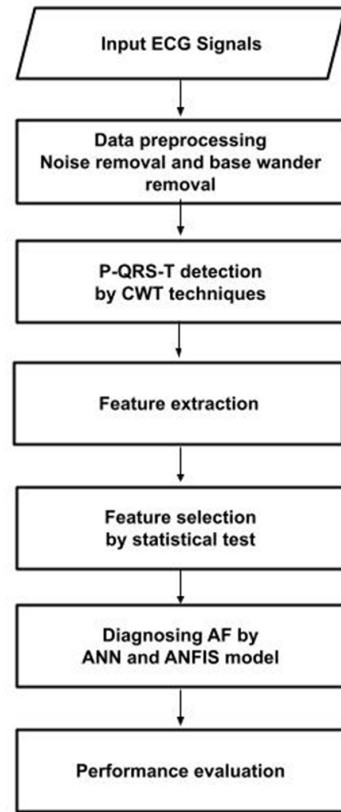

**Figure 2.** Research flow diagram.

### 2.1. Study Population and ECG Measurements

The ECG recordings of 120 samples from 60 patients used in this research included 40 NSR, 40 PAF, and 40 AF cases. All recordings were made by cardiologists in comparable conditions with patients maintained at rest and in a comfortable position. Signal recordings were acquired at a sampling rate of 1000 Hz with 12-bit resolution using lead II for a 2.5-min duration [24]. Esglhodo et al. (2022) showed the list of HRV parameters from the ECG signals of existing research works, which exhibited that the total of all signal times (number of subjects × recording duration) was between 50 min and 240 h [25]. Each sample in this research contained approximately 150 segments. In order to ensure the sufficiency of the datasets, the data augmentation technique was employed and new data sets from existing data were generated. Each group of data was combined and randomly segmented into 2.5-min intervals. So, in this research, a total of 240 samples, or 36,000 segments, were used. The total of all signal times was 600 min.

This study was approved by the Human Ethics Committee, Faculty of Medicine, Thammasat University (EC no. MTU-EC-IM-2-230/63) on 27 October 2020. The patients were individually matched concerning age (NSR: 33.85 ± 20.66, PAF: 68.43 ± 11.47, AF: 69.20 ± 9.47), gender: female (NSR: 25%, PAF: 58%, AF: 55%), congestive heart failure (NSR: 0%, PAF: 5%, AF: 10%), hypertension (NSR: 15%, PAF: 63%, AF: 65%), diabetes mellitus (NSR: 0%, PAF: 20%, AF: 25%), and vascular diseases (NSR: 0%, PAF: 5%, AF: 10%).

### 2.2. Data Preprocessing

The datasets used to determine the efficiency of this study's classification model were taken from patients' ECG recordings. Each segment had the baseline wander adjusted and noise removed with a high-pass and low-pass filter with 0.5 Hz and 50 Hz cut-off frequencies, respectively. A cycle of P, QRS, and T waves has been reported as one method to evaluate patients' cardiac condition. First, the QRS complex (the maximum amplitude) was detected based on the maximum slope. Next, the T-wave and the P-wave (the minimum amplitude and the location before the QRS complex) were detected using continuous wavelet transform (CWT) techniques [26]. Finally, the algorithm, based on the CWT in MATLAB, was used to determine each P, QRS, and T wave with a varying scale for each wave cycle, as shown in Figure 3.

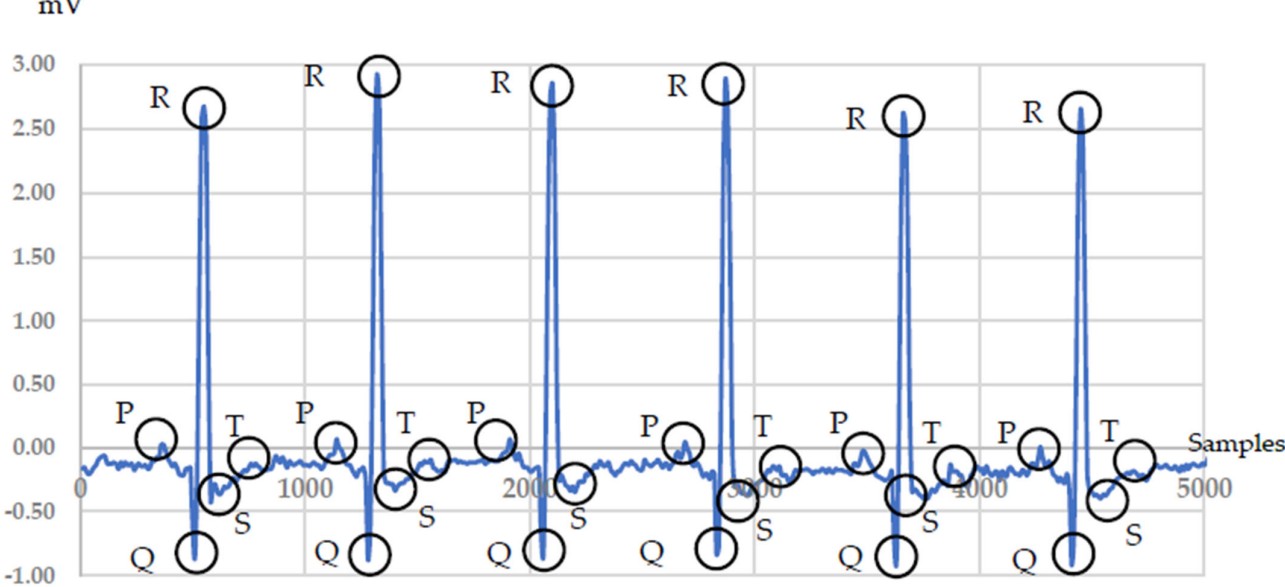

**Figure 3.** Example of P-QRS-T detection using the CWT technique.

### 2.3. Data Augmentation

A computational diagnostic model that achieves and produces good performance requires a large amount of training data to prevent overfitting problems. ECG recordings are limited in data acquisition for modeling, so time series data augmentation techniques, such as random transformation, pattern mixing, generative models, and decomposition, have commonly been applied [27]. This study selected a random transformation method to generate datasets segmented in 2.5-min intervals. Datasets were tested for sufficiency showing convergence in each epoch during training.

### 2.4. Features Extraction

The maximum value in each wave cycle is the R wave. The time interval between two consecutive R waves is called the *RR* interval [28]. *RR* intervals can be calculated to find the heart rate, for which the unit is beats per minute (BPM). AF detection can be recognized through *RR* irregularities [29]. Previous research works have detected important characteristics of ECG signals to identify AF patients, including the coefficient of variance (*CV*) [8], the root means square of the successive differences (RMSSD) [9], and the median absolute deviation (*MAD*) [10]. A new measurement, called the square of the average variation (*SAV*), is introduced in this research. These indices are used to compare the performance of AF diagnoses.

#### 2.4.1. The Coefficient of Variance (*CV*)

The *RR* interval variation was calculated from the standard deviation of the mean *RR* interval, indicating heart function with normal or abnormal beats [30]. The *RR* interval can help to assess the risk of primary cardiac arrest without clinically detecting cardiac disease [31]. *CV* is one of the typical measurements of *RR* interval variation, as shown in (1), where the standard deviation of the *RR* interval is $RR_\sigma$ and the mean *RR* interval of a group is $RR_\mu$. The *CV* value of AF patients is normally higher than that of NSR [32].

$$CV = \frac{RR_\sigma}{RR_\mu} \tag{1}$$

#### 2.4.2. The Median Absolute Deviation (*MAD*)

The median absolute deviation (*MAD*) was discovered by Hample (1974). The median is a measure of central tendency that has the advantage of being very insensitive to the occurrence of outliers. Therefore, fluctuations in the RR values relative to the median value could be used to explain the difference in AF patients [33]. The median absolute deviation (*MAD*) is a robust variable of numerical series. The *MAD* formulation is shown in (2), where the RR interval is $RR_i$, and the RR intervals within the segment are denoted as $RR_{seg}$. Linker (2016) suggested that the *MAD* of AF is a significant variable, showing a higher value for NSR than for AF [34].

$$MAD = median\left(\left|RR_i - median\left(RR_{seg}\right)\right|\right) \tag{2}$$

#### 2.4.3. The Root Mean Square of the Successive Differences (*RMSSD*)

The correlation of heart rate variability (HRV) values can be used to diagnose AF [35]. The root mean square of the successive differences (*RMSSD*) is one of the calculations of HRV that can be calculated based on fluctuating ECG signals with limited time intervals. The *RMSSD* formula is shown in (3), where the number of *RR* intervals is *N*. *RMSSD* value of AF is higher than that of NSR [36].

$$RMSSD = \sqrt{\frac{1}{N-1}\left(\sum_{i-1}^{N-1}\left((RR)_{i+1} - (RR)_i\right)^2\right)} \tag{3}$$

2.4.4. The Square of Average Variation (*SAV*)

The difference between any data and the median is a statistical measure of the number of outliers generating asymmetric data [33]. The sum of variations of the *RR* interval from the median of *RR* intervals within the segment was averaged and magnified by squaring. The square of average variation (*SAV*), as shown in (4), is presented as a new measurement technique in this research to diagnose AF.

$$SAV = \left( \frac{\sum_{i-1}^{N-1}\left( (RR)_i - median\left( RR_{seg} \right) \right)}{N - 1} \right)^2 \tag{4}$$

*2.5. ANN and ANFIS Techniques*

2.5.1. ANN

One of the classification techniques for signal processing is the artificial neural network (ANN), which computes correlations and finds solutions based on layers of neurons. ANN can find the answer through experience from training datasets [37]. It has 3 layers consisting of input, hidden, and output. The hidden layer responds to neurons from the input layer. Then, the response is delivered by generating the output vector of the output layer. Each layer has a fixed number of neuron components associated with adjustable weights. During the training process, weight values are adjusted and optimized to find the minimum error that is acceptable for use. The ANN is trained by adapting the appropriate algorithm to solve the specific problem according to the characteristic of the input data. Patterns and relationships of data are detected. The feed-forward backpropagation algorithm is commonly used in ANN [38]. The output is computed by multiplying by the weight vector and summing the results, as shown in (5), where $y$ is the output form ANN, $F$ is the activation function of $k$th neuron, $n$ is the number of neurons in the consecutive layer, $w_k$ is the weight of the respective connection of the $k$th neuron, $x_k$ is the $k$th input, and $b_k$ is the bias for the $k$th neuron. ANN has been implemented in diagnosis applications, such as computed COVID-19 detection, breast cancer, schizophrenia, and others in cardiac failure. This research diagnosed AF with different hidden layers of ANN ($n$ = 10, 15, 25, 50, and 100) that are normally used in many works [39,40].

$$y = F\left( \sum_{k=1}^{n} w_k x_k + b_k \right) \tag{5}$$

2.5.2. ANFIS

Zadeh proposed fuzzy logic in 1965 to find answers to various uncertainty problems [41]. Fuzzy logic is closely related to human thinking through differences in membership functions. The framework includes a fuzzy set, fuzzy if–then rules, and fuzzy reasoning. Combining fuzzy systems and ANN models provides a powerful tool based on learning from the ANN model and increasing performance through the fuzzy inference model. An adaptive neuro-fuzzy inference system (ANFIS) is a hybrid model integrating an adaptive ANN and a fuzzy inference system using Takagi Sugeno's Fuzzy model to define the output. The ANFIS has a mathematical structure that can detect complex non-linear systems. The ANFIS structure consists of five layers: the fuzzy layer, product layer, normalized layer, de-fuzzy layer, and total output layer, as shown in Figure 4. The ANFIS has been applied through various applications. The selection of membership functions varies depending on the characteristics of the datasets.

ECG signals are complex and have large dataset sizes. Therefore, an optimal membership function is essential to help screen AF patients. In this research, triangular (trimf), trapezoidal (trapmf), generalized bell-shaped (bellmf), gaussian (gaussmf), and gaussian combination (gauss2mf) are used in the ANFIS model to compare their performances with symmetry membership functions. These membership functions performed well in many applications and were recommended in many works [42–44].

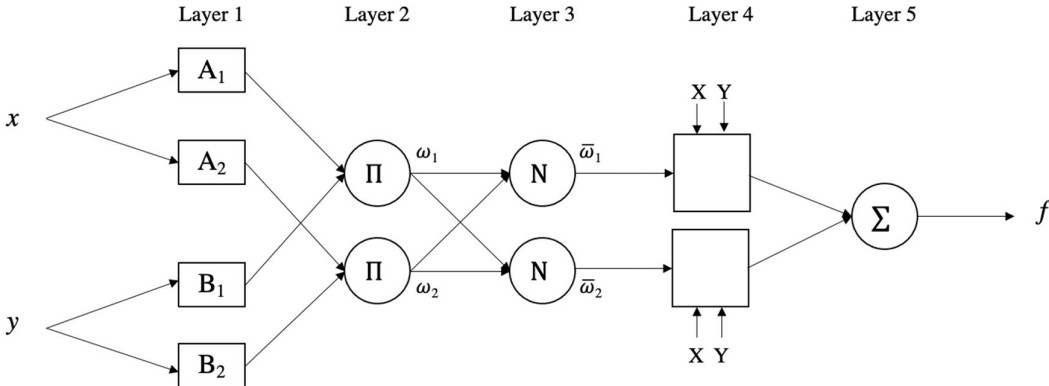

**Figure 4.** ANFIS architecture with two rules.

*2.6. Features Selection*

Screening input variables is a feature selection process for building a diagnosis model. It is used to select the significant input variables that strongly relate to the output variable. ANOVA (analysis of variance) is the discrimination tool for tests of more than two samples. Next, after the difference is found, Tukey's method is used to identify differences within the sample pairs. Minitab software was used to analyze signal data with ANOVA and Tukey's tests. The features studied in this research were CV, RMSSD, MAD, and SAV.

*2.7. Performance Evaluation*

To check the appropriate classification model before comparing ANN and ANFIS techniques, 10-fold cross-validation was used. The recorded data were randomly divided into 10 parts: nine parts taken as the training data, in turn, and the last part as the testing data. Then, cross-validation averaged the results system accuracy of 10 loop experiments to the final classification performance. The classification methods used signals with a ratio of 2:1 of training and performance testing data. Performance testing measures used to evaluate the classifiers were system accuracy (*ACC*), sensitivity (*SE*), specificity (*SP*), and positive predictivity (*PPR*), as shown in (6)–(9).

$$System\ Accuracy\ (ACC) = \frac{TP_{All\ Beats}}{n} \tag{6}$$

$$Sensitivity\ (SE) = \frac{TP_{BT}}{TP_{BT} + FN_{BT}} \tag{7}$$

$$Specificity\ (SP) = \frac{TN_{BT}}{TN_{BT} + FP_{BT}} \tag{8}$$

$$Positive\ Predictivity\ (PPR) = \frac{TP_{BT}}{TP_{BT} + FP_{BT}} \tag{9}$$

TP(AF) is the number of true positive beats when an AF beat is classified as an AF diagnosis. FN(AF) is the number of beats while an AF beat is classified as another (NSR or PAF) diagnosis. TN(AF) is the number of true negative beats when other beats are classified as another diagnosis. FP(AF) is the number of beats when other beats are classified as an AF diagnosis.

**3. Results and Discussion**

The 120 ECG recordings taken by cardiologists in a hospital and having a time interval of 2.5 min were examined. P, QRS, and T waves were detected for each segment using the CWT technique. In the feature extraction process, *RR* irregularities were transformed into variable *CV*, *MAD*, *RMSSD*, and *SAV* features. The variables from NSR, PAF, and AF patients were considered different based on mean ± standard deviation (SD), as shown

in Table 1 and the box plots in Figure 5. The mean and SD of the variables of NSR: PAF: AF for each variable were CV (0.11 ± 0.09: 0.21 ± 0.17: 0.34 ± 0.22), MAD (4.57 ± 5.76: 8.16 ± 8.75: 14.19 ± 7.78), RMSSD (54.95 ± 41.78: 81.56 ± 55.06: 109.69 ± 65.71), and SAV (4.73 ± 4.43: 22.64 ± 19.16: 253.40 ± 323.40), respectively. Heartbeat is related to the *RR* interval that has a steady rhythm under normal conditions. Previous works have shown that *RR* irregularities in AF patients have more fluctuation and are more unstable than for PAF and NSR. The results from the local patient database of those with AF illustrated higher mean ± SD values of all features than PAF, and PAF also showed higher mean ± SD values than NSR, due to the irregularity of the *RR* intervals from ECG signals.

**Table 1.** Mean ± standard deviation of variables and ANOVA test.

| Variable | *CV* | *MAD* | *RMSSD* | *SAV* |
|---|---|---|---|---|
| NSR (n = 40) | 0.11 ± 0.09 | 4.57 ± 5.76 | 54.95 ± 41.78 | 4.73 ± 4.43 |
| PAF (n = 40) | 0.21 ± 0.17 | 8.16 ± 8.75 | 81.56 ± 55.06 | 22.64 ± 19.16 |
| AF (n = 40) | 0.34 ± 0.22 | 14.19 ± 7.78 | 109.69 ± 65.71 | 253.40 ± 323.40 |
| ANOVA * (*p*-value) | 0.00 | 0.00 | 0.035 | 0.00 |

* Transformed data.

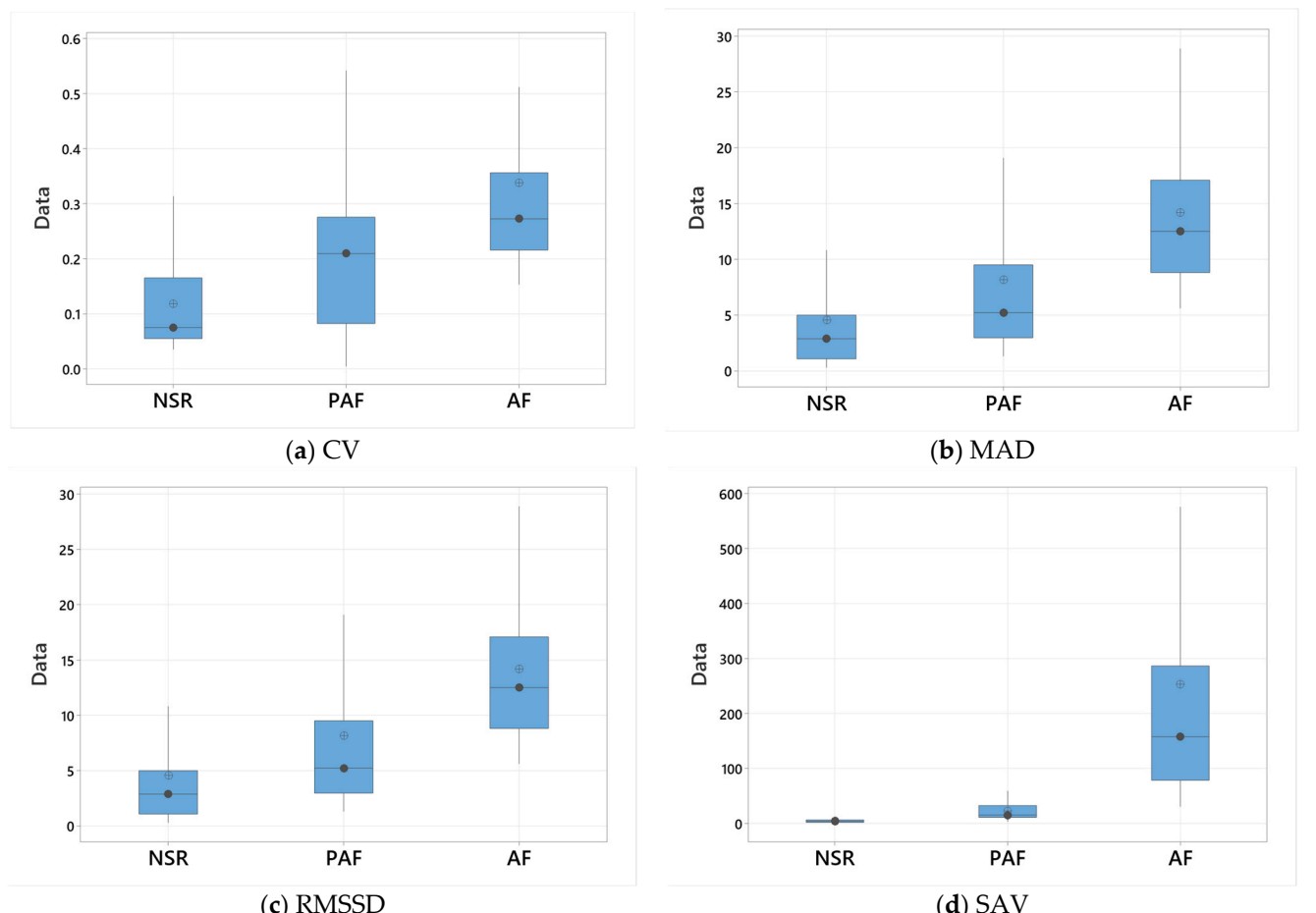

**(a)** CV        **(b)** MAD

**(c)** RMSSD        **(d)** SAV

**Figure 5.** Box plots of variables.

The ECG recordings did not fit a normal distribution. Therefore, these data had to be transformed before being used for analysis. The results from the ANOVA tests showed that all features (*CV*, *MAD*, *RMSSD*, and *SAV*) of NSR, PAF, and AF were statistically different in mean values, with a significance level of 0.05 at *p*-values less than 0.05. These variables could discriminate AF patients. From Tukey's method, all pairs of NSR, PAF, and AF were

different for all variables, as shown in Figure 6. These results have confirmed that all *CV*, *MAD*, *RMSSD*, and *SAV* of NSR, PAF, and AF can be used as discriminate indicators.

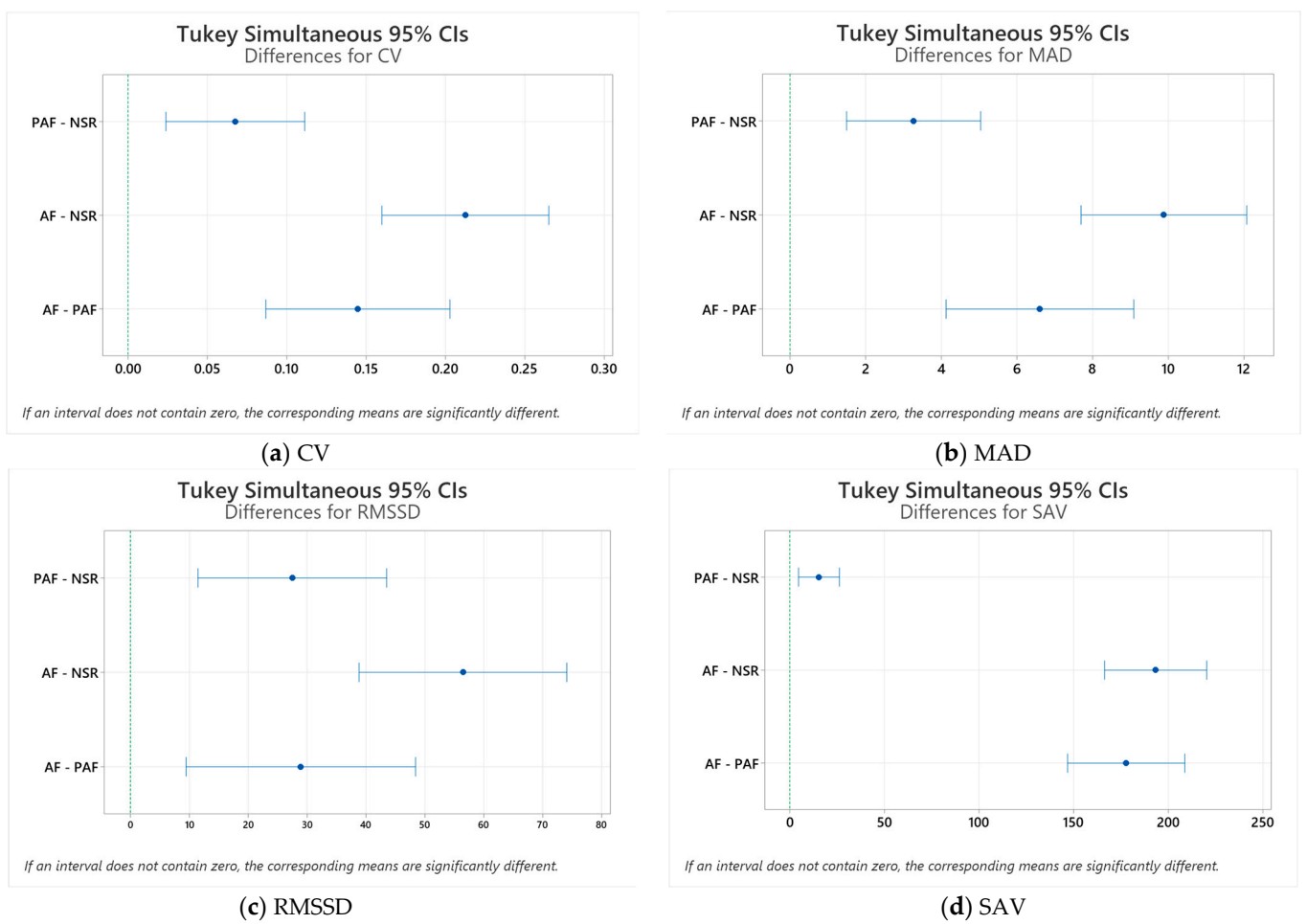

**Figure 6.** Tukey pairwise comparisons of variables, when numbers 1, 2, and 3 represent NSR, PAF, and AF, respectively.

ANN and ANFIS were selected and applied to construct the best model, and 10-fold cross-validation was assured for the AF diagnosis model. The number of neurons in ANN varied from 10 to 100. For ANFIS models, triangular, trapezoidal, bell, gauss, and gauss2 membership functions were selected for testing in this research. All types of membership were set symmetrically, with 3–5 membership functions for each variable. The results of testing models are illustrated in Table 2. This research used ACC, SE, SP, and PPR to compare the performance results of the models, which showed that the ability of the model depended on the parameters applied in the model. The performance values of the best models of ANN and ANFIS were illustrated in Figure 7. The ANN with the highest hidden layers (100 hidden layers) performed the best among ANN with different hidden layers for all features. Only ANN(50) and ANN(100) for the SAV feature could obtain ACC above 80%. ANN(100) showed very good performance for classification. The classification of AF patients is a complex problem; a large number of hidden layers is necessary for ANN models. The ANFIS with trapezoidal membership functions was the best among the other membership functions for all variables. Moreover, most ANFIS_trapmf models of each variable were better than the ANN models in AF diagnosis. The ANFIS enhances the performance of the ANN through fuzzy logic that makes the model able to capture important characteristics of the data.

**Table 2.** Summary fitting and test model.

| | %10-Fold | ACC | NSR | | | PAF | | | AF | | |
|---|---|---|---|---|---|---|---|---|---|---|---|
| | | | **SE** | **SP** | **PPR** | **SE** | **SP** | **PPR** | **SE** | **SP** | **PPR** |
| **CV** | | | | | | | | | | | |
| ANN (10) | 46.33% | 49.33% | 60.00% | 100.00% | 100.00% | 72.00% | 45.61% | 36.73% | 16.00% | 86.00% | 36.36% |
| ANN (15) | 51.31% | 53.33% | 64.00% | 100.00% | 100.00% | 76.00% | 48.21% | 39.58% | 20.00% | 88.00% | 45.45% |
| ANN (25) | 52.32% | 57.33% | 68.00% | 100.00% | 100.00% | 80.00% | 50.91% | 42.55% | 24.00% | 90.00% | 54.55% |
| ANN (50) | 69.00% | 64.00% | 68.00% | 100.00% | 100.00% | 52.00% | 75.81% | 46.43% | 72.00% | 76.00% | 60.00% |
| ANN (100) | 77.03% | 72.00% | 68.00% | 100.00% | 100.00% | 76.00% | 73.21% | 55.88% | 72.00% | 88.00% | 75.00% |
| ANFIS_trimf | 71.28% | 69.33% | 72.00% | 98.00% | 94.74% | 76.00% | 73.21% | 55.88% | 60.00% | 86.00% | 68.18% |
| ANFIS_trapmf | 71.99% | 72.00% | 72.00% | 98.00% | 94.74% | 72.00% | 78.95% | 60.00% | 72.00% | 84.00% | 69.23% |
| ANFIS_bellmf | 60.04% | 64.00% | 72.00% | 98.00% | 94.74% | 68.00% | 67.24% | 47.22% | 52.00% | 86.00% | 65.00% |
| ANFIS_gaussmf | 70.31% | 65.33% | 72.00% | 96.00% | 90.00% | 64.00% | 71.19% | 48.48% | 60.00% | 86.00% | 68.18% |
| ANFIS_gauss2mf | 67.37% | 65.33% | 72.00% | 96.00% | 90.00% | 64.00% | 71.19% | 48.48% | 60.00% | 86.00% | 68.18% |
| **MAD** | | | | | | | | | | | |
| ANN (10) | 47.34% | 49.33% | 64.00% | 96.00% | 88.89% | 32.00% | 68.66% | 27.59% | 52.00% | 70.00% | 46.43% |
| ANN (15) | 52.63% | 50.67% | 68.00% | 100.00% | 100.00% | 28.00% | 72.06% | 26.92% | 56.00% | 64.00% | 43.75% |
| ANN (25) | 57.28% | 57.33% | 76.00% | 98.00% | 95.00% | 32.00% | 77.61% | 34.78% | 64.00% | 68.00% | 50.00% |
| ANN (50) | 68.02% | 64.00% | 80.00% | 96.00% | 90.91% | 36.00% | 83.33% | 45.00% | 76.00% | 72.00% | 57.58% |
| ANN (100) | 64.36% | 65.33% | 84.00% | 100.00% | 100.00% | 28.00% | 88.24% | 46.67% | 84.00% | 64.00% | 53.85% |
| ANFIS_trimf | 66.96% | 68.00% | 92.00% | 66.00% | 57.50% | 32.00% | 89.55% | 53.33% | 80.00% | 100.00% | 100.00% |
| ANFIS_trapmf | 69.70% | 70.67% | 96.00% | 66.00% | 58.54% | 32.00% | 92.54% | 61.54% | 84.00% | 100.00% | 100.00% |
| ANFIS_bellmf | 60.32% | 65.33% | 80.00% | 72.00% | 58.82% | 40.00% | 83.08% | 47.62% | 76.00% | 98.00% | 95.00% |
| ANFIS_gaussmf | 72.66% | 66.67% | 92.00% | 66.00% | 57.50% | 24.00% | 91.30% | 50.00% | 84.00% | 96.00% | 91.30% |
| ANFIS_gauss2mf | 66.64% | 66.67% | 88.00% | 68.00% | 57.89% | 36.00% | 86.36% | 50.00% | 76.00% | 100.00% | 100.00% |
| **RMSSD** | | | | | | | | | | | |
| ANN (10) | 42.70% | 38.67% | 48.00% | 100.00% | 100.00% | 28.00% | 58.82% | 20.00% | 40.00% | 64.00% | 35.71% |
| ANN (15) | 49.99% | 44.00% | 60.00% | 100.00% | 100.00% | 52.00% | 51.61% | 30.23% | 20.00% | 76.00% | 29.41% |
| ANN (25) | 42.67% | 46.67% | 44.00% | 100.00% | 100.00% | 44.00% | 59.38% | 29.73% | 52.00% | 72.00% | 48.15% |
| ANN (50) | 51.03% | 48.00% | 60.00% | 100.00% | 100.00% | 24.00% | 71.01% | 23.08% | 60.00% | 62.00% | 44.12% |
| ANN (100) | 51.34% | 49.33% | 60.00% | 100.00% | 100.00% | 32.00% | 68.66% | 27.59% | 56.00% | 66.00% | 45.16% |
| ANFIS_trimf | 46.97% | 52.00% | 52.00% | 84.00% | 61.90% | 48.00% | 63.49% | 34.29% | 84.00% | 94.00% | 87.50% |
| ANFIS_trapmf | 68.62% | 66.67% | 92.00% | 68.00% | 58.97% | 24.00% | 91.30% | 50.00% | 56.00% | 90.00% | 73.68% |
| ANFIS_bellmf | 57.34% | 57.33% | 52.00% | 82.00% | 59.09% | 60.00% | 63.33% | 40.54% | 60.00% | 98.00% | 93.75% |
| ANFIS_gaussmf | 55.68% | 54.67% | 52.00% | 82.00% | 59.09% | 52.00% | 64.52% | 37.14% | 60.00% | 94.00% | 83.33% |
| ANFIS_gauss2mf | 52.67% | 54.67% | 56.00% | 80.00% | 58.33% | 44.00% | 68.75% | 35.48% | 44.00% | 68.75% | 35.48% |

**Table 2.** *Cont.*

| %10-Fold | ACC | NSR | | | PAF | | | AF | | |
|---|---|---|---|---|---|---|---|---|---|---|
| | | SE | SP | PPR | SE | SP | PPR | SE | SP | PPR |
| **SAV *** | | | | | | | | | | |
| ANN (10) | 60.34% | 65.33% | 80.00% | 98.00% | 95.24% | 52.00% | 77.42% | 48.15% | 64.00% | 78.00% | 59.26% |
| ANN (15) | 67.64% | 70.67% | 84.00% | 96.00% | 91.30% | 56.00% | 81.97% | 56.00% | 72.00% | 82.00% | 66.67% |
| ANN (25) | 72.70% | 74.67% | 84.00% | 100.00% | 100.00% | 64.00% | 83.05% | 61.54% | 76.00% | 82.00% | 67.86% |
| ANN (50) | 84.98% | 80.00% | 92.00% | 100.00% | 100.00% | 68.00% | 87.93% | 70.83% | 80.00% | 84.00% | 71.43% |
| ANN (100) | 86.32% | 85.33% | 96.00% | 100.00% | 100.00% | 80.00% | 89.09% | 76.92% | 80.00% | 90.00% | 80.00% |
| ANFIS_trimf | 86.32% | 85.33% | 100.00% | 92.00% | 86.21% | 76.00% | 91.07% | 79.17% | 80.00% | 100.00% | 100.00% |
| **ANFIS_trapmf** | **87.30%** | **89.33%** | **100.00%** | **94.00%** | **89.29%** | **88.00%** | **90.57%** | **81.48%** | **80.00%** | **96.00%** | **90.91%** |
| ANFIS_bellmf | 76.02% | 80.00% | 96.00% | 92.00% | 85.71% | 68.00% | 87.93% | 70.83% | 76.00% | 92.00% | 82.61% |
| ANFIS_gaussmf | 87.95% | 84.00% | 100.00% | 84.00% | 75.76% | 68.00% | 93.10% | 80.95% | 84.00% | 100.00% | 100.00% |
| ANFIS_gauss2mf | 81.00% | 84.00% | 100.00% | 86.00% | 78.13% | 72.00% | 91.23% | 78.26% | 80.00% | 100.00% | 100.00% |

* a newly proposed feature for this research.

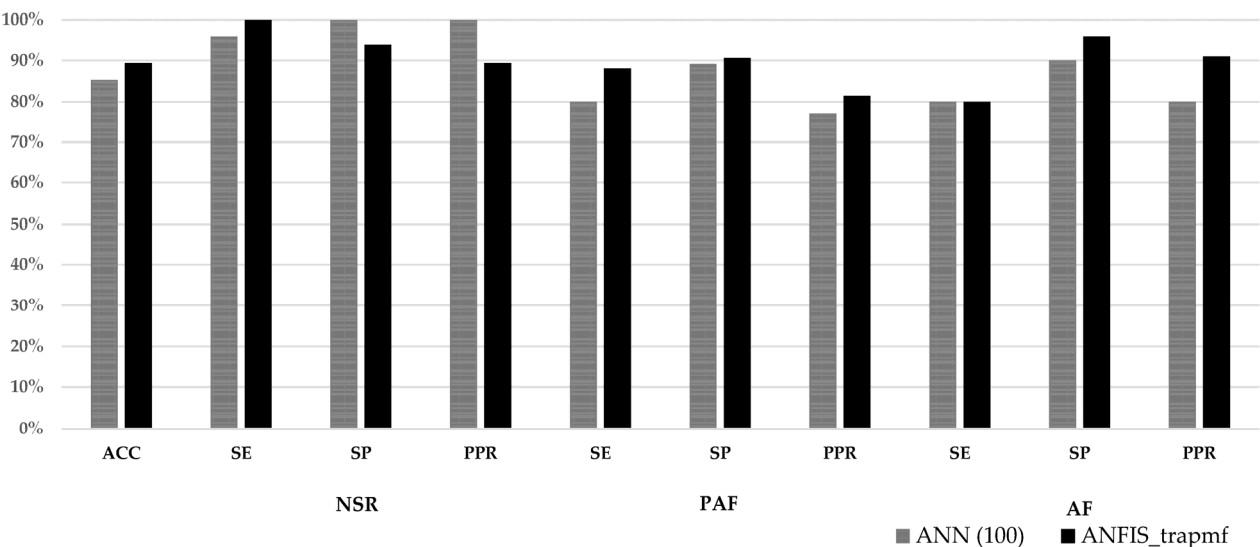

**Figure 7.** The performance values of the best models of the ANN and ANFIS.

The SAV parameter showed the highest ACC for classifying ECG recordings for both the ANN and ANFIS models at 85.33% and 89.33%, respectively. The ANN(100) showed SE:SP:PPR in NSR, PAF, and AF at 96.00%:100.00%:100.00%, 80.00%:89.09%:76.92%, and 80.00%:90.00%:80.00%, respectively. The ANFIS_trapmf were SE:SP:PPR in NSR, PAF, and AF at 100.00%:94.00%:89.29%, 88.00%:90.57%:81.48%, and 80.00%:96.00%:90.91%, respectively. The SAV is a comparison of RR intervals with the group median value. This research has improved this parameter from the MAD by magnifying the variation through squaring, so fluctuations and unstable signals can be clearly illustrated. Confusion matrixes of the SAV models for both the ANN(100) and ANFIS_trapmf models are shown in Figure 8. In the figure, the results illustrated that the ANFIS_trapmf was better than the ANN(100). In the ANFIS_trapmf, the chances of a wrong decision for NSR, PAF, and AF patients were 0%, 12%, and 20%, respectively.

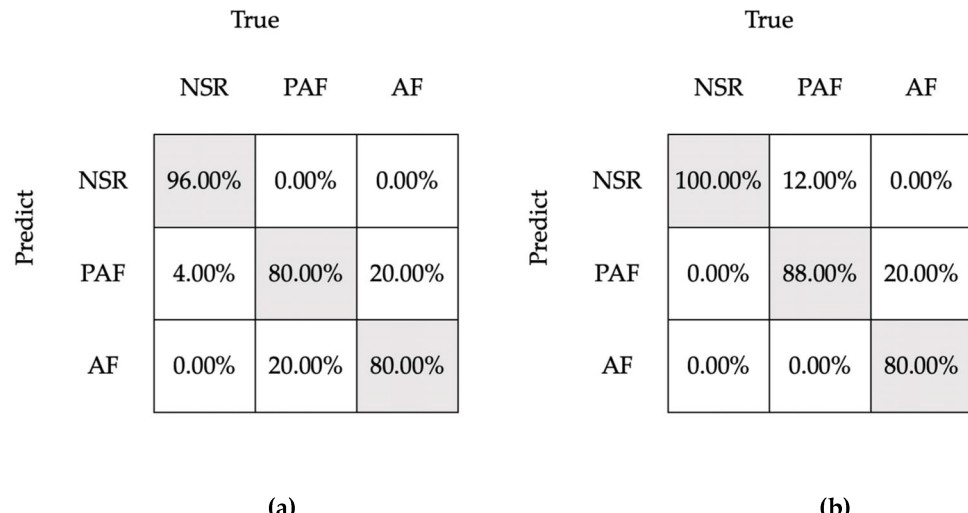

**Figure 8.** The confusion matrices show (**a**) ANN(100) and (**b**) ANFIS_trapmf models with the SAV variable.

Regarding implementation based on these datasets, the ANFIS with trapezoidal membership functions is the most suitable model for AF diagnosis, and the SAV variable is

the best indicator. It can be deployed as a decision-making tool in hospitals to assist cardiologists in diagnosing patients with AF.

Although the CNN technique has gained popularity and was used widely in AF for heart arrhythmia diagnosis, ECG recordings in this research were not abundant to train and test the CNN models. Moreover, the time interval in this research was also larger than the existing research works that used CNN in arrhythmia classifications, so the detail of the signal image was large. CNN was tested with the datasets of this research, but it could not show the performance because of insufficient datasets.

## 4. Conclusions

Diagnosis models for NSR, PAF, and AF classification using ANN and ANFIS classifiers were presented in this research. AF symptoms were determined from ECG data after collecting ECG recordings from patients in the hospital. Each segment of a signal was adjusted for baseline wander and noise was removed by a bandpass filter. The four features extracted were *CV*, *MAD*, *RMSSD*, and *SAV*. *CV*, *MAD*, and *RMSSD* are regular features used in existing classification models. However, the SAV is a new proposed variable for classification. ANOVA testing and Tukey's method were performed to screen for significant variables that would be used for the discrimination of patients. The results showed that all variables are statistically significant for AF patient discrimination. So, these features were utilized to compare the classification models of ANN and ANFIS. ACC, SE, SP, and PPR were the performance measures. ANN with 10, 15, 25, 50, and 100 hidden layers was tested, and it was found that a high number of hidden layers performed better than a low number of hidden layers. For ANFIS models, triangular, trapezoidal, generalized bell-shaped, gaussian and gaussian combinations were investigated. Trapezoidal membership performed the best for almost all types of variables. The ANFIS_trapmf performed better than the ANN. The fuzzy logic algorithm in the ANFIS could enhance the performance of AF classification. Among all variables, the SAV showed the highest performance for all methods with different parameters. The highest performance model was the ANFIS with a trapezoidal membership function at ACC 89.33%. SE:SP:PPR in NSR, PAF, and AF were 100.00%:94.00%:89.29%, 88.00%:90.57%:81.48%, and 80.00%:96.00%:90.91%, respectively. The *SAV* is the most suitable feature for NSR, PAF, and AF classification by RR irregularity features.

In future research, the best diagnosis model can be developed to diagnose AF patients. Moreover, other cardiac arrhythmias may be studied and classified using these variables. In addition, daily activity is an exciting variable that can be considered in conjunction with cardiac signals or ECG recording for investigation. The sample size in this study was relatively small, and larger sample sizes and other arrhythmias are essential to confirm this study's value. Deep learning can be applied to large datasets to compare performance.

**Author Contributions:** Conceptualization, S.D., B.P. and S.M.; methodology, S.D. and B.P.; validation, S.D. and B.P.; formal analysis, S.D. and B.P.; investigation, S.D. and B.P.; resources, B.P. and S.M.; data curation, S.D. and S.M.; writing—original draft preparation, S.D. and B.P.; writing—review and editing, S.D. and B.P.; visualization, S.D. and B.P.; supervision, B.P. and S.M.; project administration, S.D. and B.P.; funding acquisition, S.D., B.P. and S.M. All authors have read and agreed to the published version of the manuscript.

**Funding:** This research was supported financially by research funding from the Faculty of Engineering, Thammasat School of Engineering, Thammasat University Research Unit in Industrial Statistics and Operational Research, Ph.D. Scholarship from Thammasat University, and Center of Excellence in Stroke from Thammasat University, Thailand.

**Institutional Review Board Statement:** All subjects gave their informed consent for inclusion before they participated in the study. This study was approved by the Ethical Committee for Human Research, Faculty of Medicine, Thammasat University (EC no. MTU-EC-IM-2-230/63) on 27 October 2020.

**Informed Consent Statement:** Not applicable.

**Data Availability Statement:** The data presented in this study are available at doi: 10.13140/RG.2.2.25106.73921.

**Conflicts of Interest:** The authors declare no conflict of interest.

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
