# Peer review of "Comparison of ANN and ANFIS Models for AF Diagnosis Using RR Irregularities"

_applsci, doi:10.3390/app13031712_

Round 1

Reviewer 1 Report

In this research, ANN and ANFIS models for AF diagnosis using RR irregularities are applied. The square of average variation (SAV) - a newly proposed feature that extracts from the irregularity of RR intervals – is proposed. This research collected ECG signals from local Thai patients to create a classification model. This is a plus point. The methodology is well-explained, and simulation results support the concept. Relevant references are provided.

However, some major concerns need to be addressed.

1.       Only the ECG recordings of 120 samples are used in this research. Is this data enough to train ANN and ANFIS? Is there any possibility to take more samples to train ANN and ANFIS?

2.       The authors are advised to share data (make online) for the benefit of researchers.

3.       The ECG recordings of 120 samples used in this research included 40 NSR, 40 PAF, 114, and 40 AF cases. It has been observed that extensive preprocessing has been done to select 40 NSR, 40 PAF, and 40 AF cases to avoid bias/oversampling/data augmentation. This is not recommended. The developed program should be able to select these cases based on data augmentation techniques.

4.       Due to the reduced number of samples, extensive computations have to be performed by varying parameters. This process is quite laborious; How can you justify this?

5.       ANN and ANFIS are quite traditional methods now. Here they have applied to self-recorded ECG data. How can you justify novelty and creativity?

6.       Comparative analysis with other techniques especially deep learning techniques is missing. 

Author Response

Dear Reviewer,

Thank you very much for your valuable comments. All comments have been checked and answered in the attachment.

Best Regards,

Busaba Phruksapharat

Reviewer 2 Report

Research summary: 

The topic is relevant and may be of interest to a broad range of the journal's readers. However, this reviewer has some major concerns about the paper.

Major Strengths: The major strengths of the research are:

- The topic is interesting

- The proposed approach has been properly designed and developed

Major Weaknesses: The major weaknesses of the research are:

- The model results are not optimal

- The structure and contents of the paper need to be improved.

- The evaluation is weak 

Grammar and Readability:

The paper is well-written and clear. I didn't find any typos.

Specific Comments: My specific comments concerning this manuscript are:

- The abstract does not highlight the specifics of the research or findings but contains too much background information. Some details of the research would be nice for example numbers addressing the sample, data, percentage improvement, etc.. Remove some of the background material and add some details of the research. Moreover, it is good to provide some specifics (e.g., sample size, dataset size, numbers from results, etc.).   

- There needs to be an explicit research objective(s) and/or research question(s) stated, preferably as a separate section. This helps readers find out what the research is trying to address.

- There are several papers that have addressed similar problems, but it is necessary to further highlight the novelty between the proposed study and the related literature.

- A related work section is missing.

- Please add recent references. Certainly, there has been more recent (within the last two years) research on this topic published in information science and/or computer science outlets. An academic search on the topic (using keywords from the manuscript’s title) shows that there is recent work in this area. I suggest some recent works start with: https://doi.org/10.3390/sym14061139, https://doi.org/10.18293/DMSVIVA21-008, https://doi.org/10.1109/IEC54822.2022.9807511

- The reference list needs tidying up, as there are references missing items or formatting issues. Please be consistent with the formatting and use some standard formatting style.

- Subfigures 1(a), (b), and (c): a caption is missing 

- The quality of Figures is quite low and often does not fit with the template of the paper.

- Please consider reporting the results of Table 2 as one or multiple plots

- The discussion of the results should be more detailed in order to highlight the best configurations. Furthermore, the sections of the results lack detailed reasons for the findings. It appears that the authors have launched the models and reported the results.

Concluding Remarks:

The goal of the paper is interesting. I think that the paper should be further improved according to the considerations I reported in the review. Moreover, the authors should better highlight the novelty of the paper with respect to the SOTA.

Author Response

(The authors gave the same response as above.)

Round 2

Reviewer 1 Report

Major concerns have been addressed.

 Some minor suggestions are:

1.       Perhaps the share data (make online) is now working properly. Please check

 2.       Since the data augmentation technique was applied and new datasets from existing data were generated, the authors are advised to add a separate section/subsection for the data augmentation technique used in the manuscript.

 Good luck!

Author Response

Dear Reviewer,

According to your comments. We have already shared data in ResearchGate and added a section about data augmentation (section 2.3).

Thank you for your valuable comments.

Best Regards,

Busaba Phruksaphanrat